# Effects of Using Green Concrete Materials on the $CO_2$ Emissions of the Residential Building Sector in Egypt

Heba Marey [1,*], Gábor Kozma [2] and György Szabó [1]

1  Department of Landscape Protection and Environmental Geography, Institute of Earth Sciences, Faculty of Science and Technology, University of Debrecen, 4032 Debrecen, Hungary; szabo.gyorgy@science.unideb.hu

2  Department of Socio-Geography and Regional Development, Institute of Earth Sciences, Faculty of Science and Technology, University of Debrecen, 4032 Debrecen, Hungary; kozma.gabor@science.unideb.hu

*  Correspondence: heba.marey@science.unideb.hu

**Abstract:** Increasing the rate of construction material consumption has caused significant environmental problems in recent decades, especially the production of ordinary Portland cement (OPC), which has been associated with 8% of the world's human $CO_2$ emissions and is considered the leading binder of concrete. This study aims to investigate the effects of substituting conventional concrete (CC) material with green concrete (GC) in the non-structural concrete works of a residential building in New Borg El-Arab City, Egypt. It attempts to establish what the effects are of using GC on cement, natural aggregates, and $CO_2$ emissions in the design phase. By using a design-based solution (DBS), we began with redesign, reduce, reselect, reuse, and recycle strategies to find an optimal solution for applying recycle aggregate concrete (RAC) as a replacement material in selected building parts, such as the internal floor, external sidewalk, entrance steps, and wall boundary. AutoCAD software and 3Dmax were used to modify the original design and obtain two design references with four different scenarios. Comparative analyses were applied to investigate the effects of different concrete materials. The results show a reduction of about 19.4% in cement consumption in terms of the total concrete of the building and a 44.5% reduction in $CO_2$ emissions due to the reduction of cement in specific building parts. In addition, this solution decreased natural coarse aggregate (NCA) consumption by 23.7% in the final concrete. This study recommends that GC materials close the loop of cementitious material consumption to reduce environmental impacts and achieve sustainability in the Egyptian building sector.

**Keywords:** green concrete; $CO_2$ emissions; residential building; Egypt

## 1. Introduction

$CO_2$ emissions and the depletion of natural resources are associated with the consumption rate of construction materials in the building sector worldwide [1]. For example, cement consumption in the United States reached 102 million metric tons in 2020 and was strongly linked with construction industry demands [2]. The main challenge in this context is the high consumption of concrete, which represents a substantial problem with regard to cement production, which contributes around 8% of the world's carbon emissions, and this is reflected in the rate of cement production, which increased from 1.39 billion tons in the year 1995 to 4.1 billion tons in the year 2020 (Figure 1). As a result, by 2010, cement accounted for 36% of the 7.7 Gt $CO_2$ emissions from the construction industry [3]. Consequently, concrete is considered the second largest producer of $CO_2$ emissions globally, and is considered one of the main reasons for the global demand for construction aggregates, which exceeds 26.8 billion tons per year.

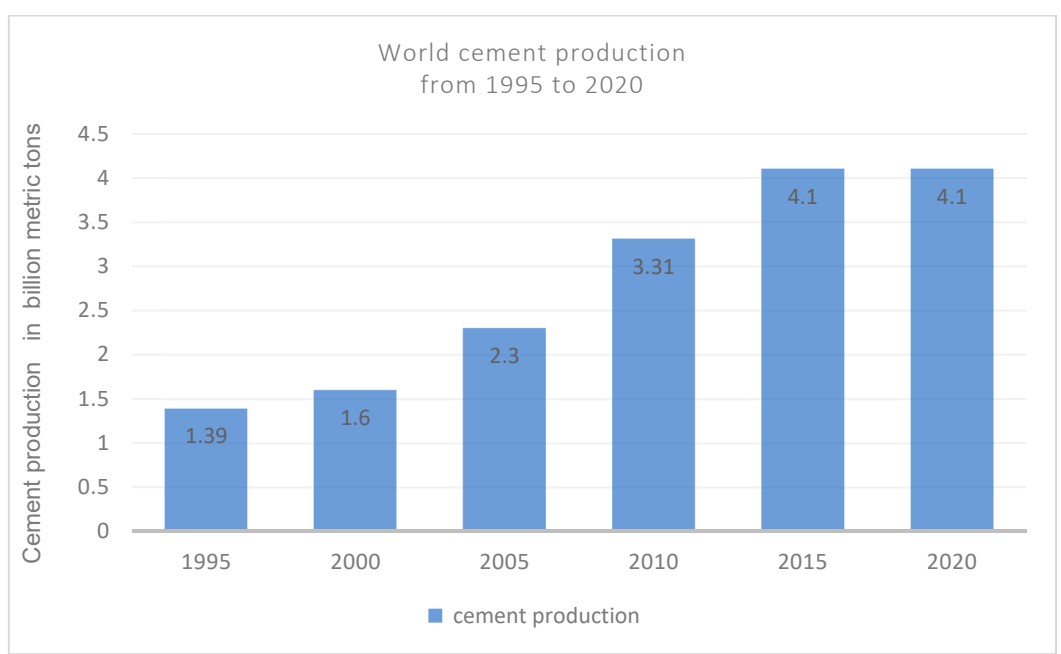

**Figure 1.** Global cement production from 1995 to 2020, in billion metric tons [4].

Concrete is defined as a composite material formed from cement, aggregate, and water, and cement is considered a binder that can bind mixed materials together to produce concrete composite, and so each ton of OPC releases approximately a similar amount of $CO_2$ into the atmosphere [5]. Concrete has different characteristics depending on its ingredients, using relative quantities and specific amounts of water. Therefore, it is the world's most significant construction and architectural material, because it has a formative ability to change its components, site preparation method, and curing process. In the same context, the components comprise four main ingredients: water, Portland cement, aggregates, and air, consisting of two parts (aggregates and paste). Aggregates are divided into fine and coarse, and account for 60 to 80 percent of concrete. The paste comprises cement water and entrained air, and usually accounts for 20% to 40% of the overall composite amount [6]. However, in 2006, concrete production reached 7.5 billion cubic meters per year, more than one cubic meter for each person on Earth. Furthermore, the demand for construction materials and natural aggregates exceeds 26.8 billion tons/year [3], and is expected to increase in the next few years, resulting in severe negative impacts on the environment, such as $CO_2$ emissions, dust, particulate matter, water pollution, primary energy use, and the depletion of non-renewable resources [5–8]. Consequently, the need for innovative construction materials has become essential, especially in the building sector [9].

### 1.1. Green Concrete

GC has many sustainable features that effectively replace CC. For example, it is defined as concrete containing recycled materials as one of its components, or concrete whose manufacturing process does not harm the environment and has high performance during a building's life cycle [9–12]. Furthermore, GC materials have an essential role to play in reducing the environmental impact of construction activities due to their ability to use advanced technology and alternative environmentally friendly materials [13,14] as well as reduce $CO_2$ emissions, energy use, and non-renewable resource consumption. Various scholars worldwide have worked to improve the strength, durability, workability, and fire resistance of GC by improving strategies based on recycled materials, optimizing design properties, and reducing emissions based on replacing Portland cement (PC) with alternative by-product cementitious materials [14,15]. Hence, GC is expected to demonstrate excellent hardened properties, reduce maintenance and construction costs, and improve the service life of buildings, thereby reducing negative environmental impacts and achieving a

circular economy [9,10]. On the other hand, concrete's compressive strength is dependent on many factors, such as the component ratio, type of cement, source of aggregates, grain sizes, and properties to determine the behavior of concrete. Generally, GC is a term usually used for environmentally conscious or friendly concrete materials, with the inclusion of recycled wastes and environmentally friendly materials such as fly ash, silica fume, and construction and demolition (C&D) waste, which can be used as eco-friendly materials as a portion of the GC cement mixture. Accordingly, GC has the potential to meet the environmental and economic objectives of sustainable development [16,17].

## 1.2. Cement and $CO_2$ Emissions

Cement's ingredients are available worldwide; consequently, cement production has increased rapidly in recent years. OPC is the most common contemporary cement used in construction activities, and its production strongly contributes to anthropogenic $CO_2$ emissions. Nowadays, global production has reached more than four times that seen in 1990, and 73% of global cement production occurs in China. Therefore, cement production is considered the third largest producer of carbon dioxide, after fossil fuel consumption and land-use change, and the estimation system of $CO_2$ emissions from fossil fuels and industry ($E_{FF}$) is based on energy consumption and cement production [18–20].

The process with the highest $CO_2$ emissions is the production of components; clinker (carbonate limestone and $CaCO_3$) is decomposed into oxides, lime, and CaO, and $CO_2$ is released by heating, with this process being associated with about 5% of total anthropogenic $CO_2$ emissions. On the other hand, the massive amounts of fossil fuels that are used to generate the required heat for the raw materials, reaching over 1000 °C, are referred to as "energy" emissions and add a further 60% to the overall emissions (IEA, 2016 [18,19]). The emissions from the industrial cement process represent approximately 8% of global $CO_2$ emissions [21]. Consequently, the production of one ton of PC clinker creates approximately one ton of $CO_2$ and other greenhouse gases [18]. Furthermore, massive environmental impacts are caused during cement production due to the high-temperature calcination of carbonate minerals, causing clinker and $CO_2$ to be emitted into the atmosphere [22,23]. Although PC production is considered a critical factor in the context of achieving sustainability, the related relationship between GC, cement production, and $CO_2$ emissions needs further investigation, as do innovative solutions based on cooperation between architects, engineers, stakeholders, cement manufacturers, and the public and private construction sectors [14,24,25].

## 1.3. Recycled Aggregate Concrete (RAC)

Various studies have compared different materials in buildings and positively reported that GC is a more environmentally friendly material and has the same functionalities and properties as CC in terms of building life service, with low costs and a reduction in negative impacts. In the same context, Tam et al. [26] discussed the utilization of C&D waste as RA for infrastructure development, which depended on the specification and quality of the waste. They mentioned that RAC could enhance and achieve environmental and economic benefits by reducing $CO_2$ emissions and promoting civil engineering works, such as concrete pavements and roadways, and recommended using RA materials on a larger scale in construction projects. However, the characteristics and applications of RCA need development. Al-Mansour et al. [27] discussed using RCA in concrete mixes to replace NCA. They found that it can reduce the environmental impact and the cost of NA production, but proper treatment should be applied to control the particle size distribution and maintain the balance between resistance and concrete compressive strength in order to enhance concrete quality. The mechanical properties of concrete depend on RCA percentages, and so the substitution ratios of 25% and 100% reduced the mechanical strength of the concrete by 5% and 25%, respectively, and there was no significant effect. They recommended using the materials for general purposes and moderate conditions. In the same regard, El-Hawary et al. [28] examined the quality of recycled concrete and

the usage of RCA as a total replacement for NCA, and they found that the mechanical properties of the concrete products increased to a strength of 35 MPa with a fixed water to cement ratio of 0.48 for all the concrete mixes [29]. Furthermore, the study by Rahal K. [30] compared the mechanical properties of RAC made from demolished concrete structures with NAC and examined ten concrete mixes using regular or RCA with different strengths ranging from 20 to 50 MPa. The results show that the 28-day cube and cylinder compressive strength and the indirect shear strength of RAC were, on average, 90% of those of NAC with the same mix proportions, and that they played a similar role. On the other hand, the modulus of elasticity of RAC is lower than NAC by only 3% between compressive strengths of 25 and 30 MPa, and the strain at peak stress in RAC is similar to NAC.

However, Al-Gahtani et al. [31] examined the stainability benefits of using demolition and industrial wastes as replacement materials for aggregates and cement, respectively, by using crushed demolition site waste as a fine and coarse aggregate in concrete to produce GC. This study showed the adequacy of using such recycled demolition materials as replacements in the production of GC without significantly changing its mechanical characteristics, enhancing its properties, or increasing pozzolanic industrial refuse and by-products.

In the case of the Egyptian construction sector, this is considered an effective solution for achieving sustainability, supporting Egypt's vision for 2030, which is concerned with environmental protection and the conservation of natural resources [31,32]. Wagih et al. [2] discussed the possibility of replacing NCA with RCA in structural concrete at different proportions. The results show that it could be utilized as a recycled aggregate and is suitable for concrete structural applications in Egypt. They found that using RCA materials in Egypt has a positive impact on the construction industry and can reduce quantities of C&D waste that causes negative environmental impacts. From the same perspective, Sharkawi et al. [33] explored the use of RCA, and most of the concrete mixes exceeded 17 MPa as non-reinforced concrete, which is used in infrastructure works such as pavement and sidewalks to reduce negative environmental impacts, and the reduction in concrete compressive strength from 37% to 62% depended on the opportunities for replacing the NCA with recycled concrete produced from C&D waste. They prepared C&D samples collected from different demolition sites and disposal locations in Tanta, Egypt. The results show that C&D has a solid ability to transform the type of C&D constituents.

*1.4. Residential Building Materials*

The increasing construction activities of the residential building sector are considered the largest consumer of raw materials and the largest generator of waste materials; thus, GC offers an essential opportunity to reduce construction waste, the consumption of non-renewable resources, and $CO_2$ emissions [31,34]. Therefore, residential building architectural design is responsible for selecting the proper GC materials, especially in the first design stage, and is able to reduce CC consumption through sustainable environmental and economic considerations. From the same point or view, Baldwin et al. [35] focused on the importance of adapting the design phase and cooperating with construction project experts to achieve reduced waste in high residential buildings. They investigated the effect of using design structure matrix techniques that helped to reduce the amount of concrete materials used during the construction and design process. The study analyzed the designer's role in understanding the criteria and implications of different innovative concrete methods within the early design stage and proved that the designer has an essential role to play in reducing material waste.

One of the most significant obstacles facing the construction process is selecting appropriate and efficient methods of recycling concreting materials in C&D for the purpose of protecting the environment and promoting sustainable development strategies. RCA has a crucial role to play in achieving the recycle-closed-loop strategy in C&D waste management and the ability to achieve a greener environment [21]. In this regard, economic benefits were addressed by Kumar et al. [36], who used and analyzed a cost–benefit

model of RAC production and focused on the economic effects. This study found that GC materials produced from RA that are used in industrial-scale manufacturing have low prices. Furthermore, achieving the benefits of recycling C&D waste closed the loop of raw material consumption.

In the same context, the study by Mi et al. [37] explored the influence of the strength of the two materials, original concrete and RAC, on compressive strength and carbonation resistance. The result recommended that construction waste treatment organizations classify waste concretes from different sites rather than mixing them, and then directly utilize them in actual projects. Moreover, using sustainable and green materials in rural Egyptian communities was investigated in the study of Ali et al. [38], which compared the operational processes of traditional and modern building materials and investigated the influence of two types of building materials on environmental impacts, aiming to recognize modern techniques for sustainable building construction in rural communities in Egypt.

Additionally, Khalil and Abouzeid [39] investigated the importance of selecting appropriate building materials and using different wall materials commonly used in the Egyptian construction sector, such as autoclaved aerated concrete (AAC) and red bricks, to address environmental issues and the final project costs. They applied a comparative analysis of the most common bricks and wall systems and the utilization of techniques in the construction sector in the Egyptian Administrative Capital as a case study of more efficient and sustainable buildings materials.

In the Egyptian construction sector, in which environmental protection is a significant concern, demolition debris has increased and is considered a viable key material for recyclable material solutions used in road applications. The study by Behiry et al. [40] examined the feasibility of RCA mixed with limestone aggregate, which is used as a base or subbase application in Egyptian roads. The results show that adding RCA improves the mechanical properties of the mixture, where the unconfined compressive strength is taken as an essential quality indicator. Variables influencing strength, such as cement content, curing time, and dryness, play an important role in determining the performance of treatment with RCA. However, the study by Naik and Moriconi [41] considered that concrete made from recycled materials is a sustainable material, due to the fact that it has very low inherent energy requirements, can be produced to order as needed, and can be made from recycled materials. Adding more high-performance cement can reduce the amount of cementitious materials by applying supplementary materials instead of cement in the concrete component. This study recommended that $CO_2$ emissions be reduced by expanding GC production. By using an integrated strategy of sustainable design and construction for the structures of residential buildings, the impact on the environment can be reduced to the minimum level. The study by Sadek, D.M. [42] investigated the strength and durability of high-performance concrete made with air-cooled recycling slag (ACS) as a replacement for NCA by assessing the advantage of using ACS for concrete containing RCA, namely, that it mitigated the harmful effects of RCA on strength and longevity without increasing the cement content. The study concluded that it is feasible to produce high-performance concrete with satisfactory properties using RA and supplementary cementing material and that the resulting product performed better than concrete made with natural aggregates.

## 1.5. Research Objectives

This study aimed to investigate the effects of substituting CC material with GC by identifying the influence of modifying the original building design and the impact of using RAC materials on the consumption of raw materials (sand and NCA) and $CO_2$ emissions in the Egyptian residential building sector. This study attempts to answer the following question: what are the effects of using GC instead of CC on cement consumption, natural aggregate, and $CO_2$ emissions in residential buildings? Moreover, further questions are defined as follows:

-    How can the modified design play an essential role in concrete material consumption?

- What is the impact of GC on the consumption of cement and natural aggregate?
- What is the impact of using RAC on $CO_2$ emissions?

The research goal was to identify the efficiency of using GC in residential buildings and guide the decision-making of stakeholders in the construction industry.

However, the green transformation of the construction industry is a significant issue, and the reduction in CC use is the most important goal in this context. Therefore, this study promotes a sustainable strategy for the application of GC materials and demonstrates that substantial decreases in conventional construction materials can be achieved. Although several previous studies have discussed the properties, strength, and durability of GC materials, there are limitations to the empirical studies concerned with applications to specific building parts in non-structural works in the residential building sector. This research represents the first attempt to apply green concrete and presents the first sustainable building model using GC materials in the residential building sector in Egypt. The flexibility and ability to compare different residential case studies locally and globally provide opportunities for more investigations and research efforts in the future.

## 2. Research Methodology

The design-based solution (DBS) technique was applied as an integrated method to investigate the role of concept design performance and GC material selection on natural resource consumption and $CO_2$ emissions when using RAC made with RCA from C&D waste in the Egyptian residential building sector.

We classified the concrete materials into three groups

1. C1—reinforced concrete with CC.
2. C2—non-reinforced concrete with CC.
3. C3—non-reinforced with GC.

The research applied strategies related to redesign, reselecting materials, replacing conventional materials, recycling C&D waste materials, reusing RAC, and reducing the environmental impacts of concrete consumption in the following case.

First, we began with redesigning and re-selecting strategies by reviewing the original design of the existing building, using AutoCAD software to develop a modified design and obtain two reference designs (original and modified) with the subcategories A and B as follows:

Original A: original design with CC materials.
Original B: original design with GC materials.
Modified A: modified design with CC materials.
Modified B: modified design with GC materials.

The modified design was applied to assess whether it was suitable for applying GC materials for non-structural concrete works in the selected building parts in order to obtain hypothetical options with the same basic design features and equivalent functionality and durability levels.

Second, in context of the replacement of CC with GC made from RCA, we calculated the concrete quantities after modification for specific building parts and identified corresponding components such as cement, sand, natural aggregate, and recycled aggregate based on the bill of quantity documents (BOQ), feasibility project study, and official contractor documents. The absolute volume method was used to calculate the mix proportions of concrete compositions using Microsoft Excel, considering [2].

Third, comparative analysis was applied to assess the concrete quantities of the building references after the replacement of CC with GC, using SPSS and jamovi software to identify the greatest reduction in cement material consumption.

Fourth, we identified the effect of the substitution of GC materials in original and modified buildings on the $CO_2$ emissions from the production of cement (OPC) materials. Then, we achieved estimates for buildings based on the CDIAC's method for estimating the emissions from cement production reported by [18,33,43].

In this research, early estimates of $CO_2$ emissions from cement production assumed that cement was of OPC type and calculated the cement quantity per ton. Consequently, the CDIAC's method estimated the emissions arising from cement production [43,44]. In addition, we assumed that $CO_2$ emissions occurred during cement production when converting the limestone to lime and producing clinker, and so the calculation was as follows:

$$CO_2 \text{ (as C)} = P \text{ Cl C}$$

where P is cement production, Cl is the clinker fraction, and C is the carbon as $CO_2$ per product based on the review of the previous studies. The report by Griffin (1987) (Andrew, R. M 2018) [18,43] required that cement production data in tons are multiplied by a fixed factor of 0.136 to obtain tons of carbon emitted as $CO_2$, i.e., 1 ton of cement produced $0.136 \times 3.667 = 0.50$ tons of $CO_2$ [3,4,18,19,24,43,45].

Finally, a conceptual framework was developed for using GC materials for the residential building sector in Egypt as a guideline for stakeholders and the construction industry. It provided an optimum building model using 3Dmax software to present a clear final image of the modification criteria and acts as a guideline for future residential projects.

### 2.1. Description of Egyptian Residential Building Case Study

### 2.1.1. The Building Data and Information

This research work was carried out on an actual residential building project in the design phase. The modified design presented a hypothetical option for standard sustainable construction materials. The building considered was a contemporary residential villa, and its documents met the Egyptian Building Code requirements. The concept design phase is concerned with a specific part of the building, which is designed to consider contractor documents and government building regulations. The BOQ is considered a reference for construction works. The building drawings illustrated the part's dimensions and details. However, the data in the current research are concerned with architectural and structural drawings, and mechanical, electrical, and plumbing (MEP) works are not within this study's scope.

The selection of this case study was based on it being considered the most common building typology in new Egyptian cities under the government's development strategy, the aim of which is to increase the amount of family housing in order to address the population growth crisis and improve urban quality of life. This can also be applied in many other countries with similar building and concrete material code considerations. The flexibility of applying GC materials in such a building type will encourage construction companies, architects, engineers, and the concrete industry to apply sustainable construction materials.

### 2.1.2. Data Collection

Building data were collected from the official main contractor's project documents, confirmed by a consultant project team made up of architects, civil engineers, and quantity surveyors, and supported by official project requirements and reports, the original project plan, building information data, a site visit, in situ measurements, and official government documents based on the Egyptian building code and building authority regulations [2,33]. The data were validated and signed by a third-party consultant team to match Egyptian building code specifications and construction regulations. Furthermore, the data were considered part of project cycles and could be precisely and reliably applied for each selected building part. Figure 2 shows the building located in the urban scope of New Borg El-Arab City, Alexandria Governorate, Egypt, which is a standard family building class within the government's development program.

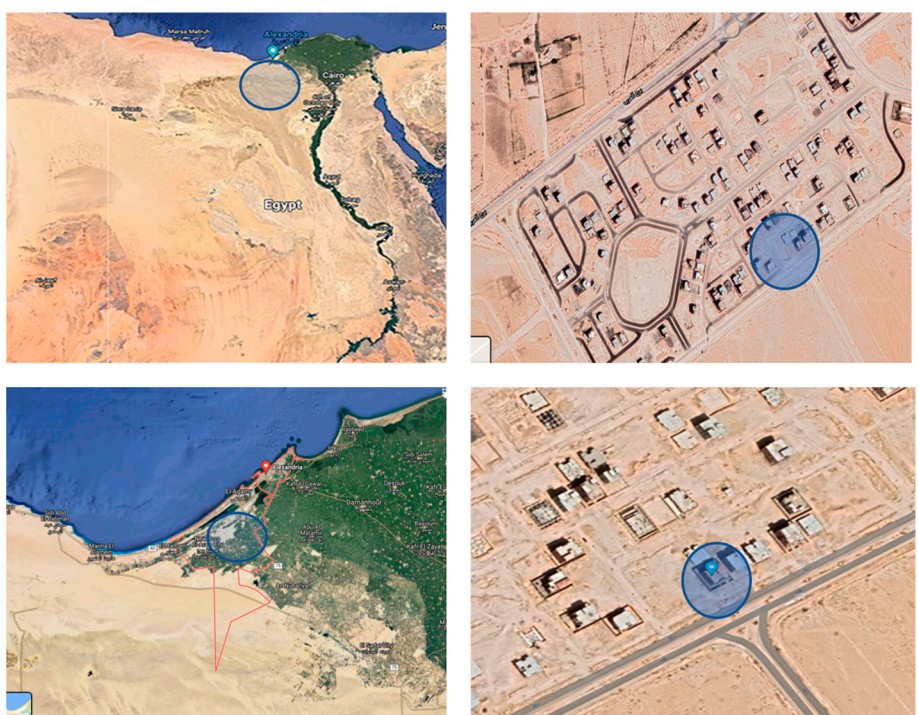

**Figure 2.** The building's location in New Borg El-Arab City in Alexandria, Egypt, source Google Map [46]. Imagery ©2022 CNES/Airbus, Landsat/Copernicus, Maxar Technologies, Map data ©2022.

## 3. Results

### 3.1. Original and Modified Building Specifications

In this study, the modified building design considers all the essential building elements of the original design and their corresponding parts. The modified design maintains the equivalent land use and building dimensions. Consequently, it consists of a basement, a ground floor, a first floor, and a penthouse, in addition to a surrounding garden with a total land area 500 m², and the total built area is 666 m², with a total height of 14.1 m. The basement area is 192.8 m², with a height of 3 m, and the ground floor area is 212.1 m², with a height of 4 m. The first floor has an area of 212.1 m², with a height of 3.5 m. Meanwhile, the penthouse is 49.008 m² in area, with a height of 3 m due to building regulations. Table 1 presents basic information regarding land use and building dimensions depending on the technical specification report and contractor documents, considering sizes, elements, shapes, and forms.

**Table 1.** Basic information regarding land use and building dimensions.

| Building Part | Dimensions | | | Floor Space Area, m² |
|---|---|---|---|---|
| | Length, m | Width, m | Height, m | Area, m² |
| Basement | 14.39 | 13.40 | 13.50 | 192.81 |
| Ground floor | 15.54 | 13.65 | 4.00 | 212.09 |
| First floor | 15.54 | 13.65 | 3.50 | 212.09 |
| Penthouse | 6.13 | 8.00 | 3.00 | 49.01 |
| Total build area | | | | 666.00 |
| Total land area | 20.00 | 25.00 | | 500.00 |

Modifying the Original Building Design

The modified design was applied to residential buildings by modifying the original design. In this current study, specific building parts were selected to be suitable for GC

implementation. The GC selection parameters included recyclable quality, the availability of materials, and the equivalent durability, strength, and functionality, and RAC produced from local C&D waste was selected to reduce the consumption of concrete materials and natural resources and limit $CO_2$ emissions. The modification of the internal building space was applied in the basement by removing the extra decorative internal staircase and adapting the lift space, and the internal space area of the ground floor was also increased by reducing the number of unnecessary walls and staircases. Figure 3 shows the front elevation, typical site plan, and the floorplans for the basement, ground floor, first floor, and penthouse for the original and modified designs. Furthermore, the concept design is concerned with improving the aesthetics of the building's architectural design and modifying its style, for example, increasing the decorative motifs on the façade wall and the roof design. Moreover, the design improves external decorative works, such as garden furniture, the fountain, pavement, sidewalks, the wall boundary, and entrance steps. However, the concept design, which was prepared to meet the requirements of sustainable strategies for material efficiency is concerned with the standard criteria of energy efficiency strategies in the construction phase. Therefore, the concept design presents the first sustainable building model using GC materials in an urban context.

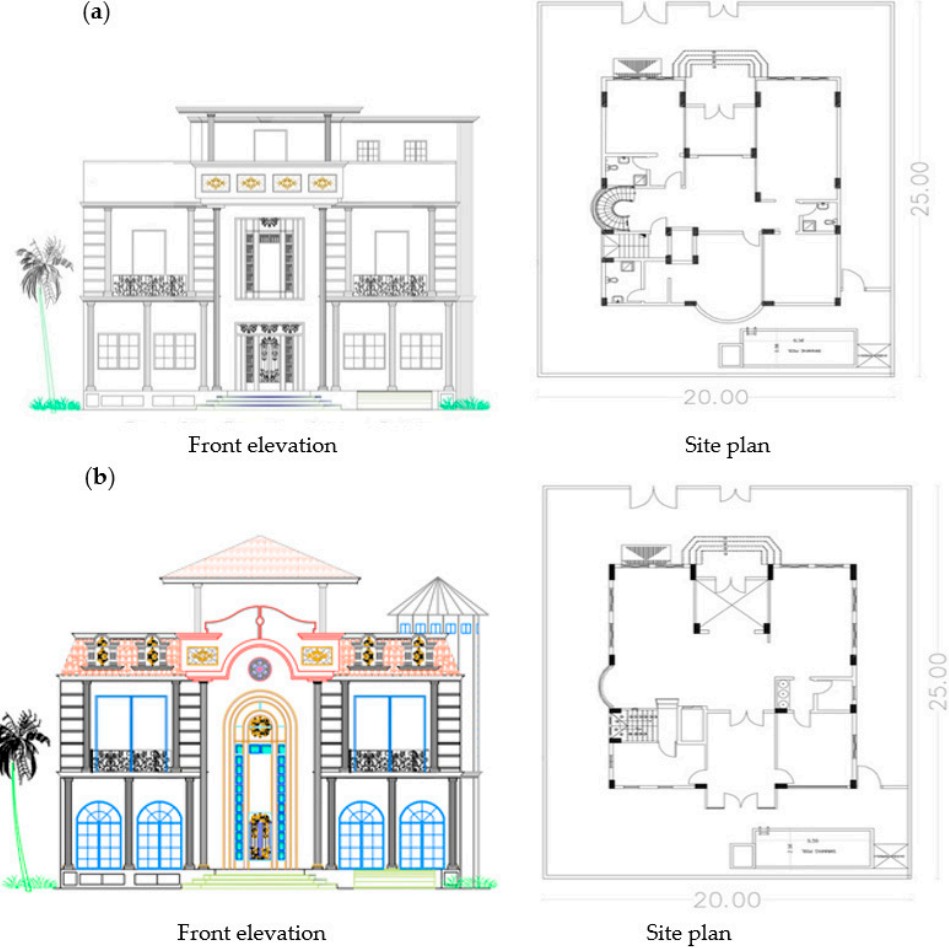

**Figure 3.** (**a**) Original design front elevation (left) and the typical site plan (right). (**b**) Modified design front elevation (left) and the typical site plan (right) (source: authors).

### 3.2. Selecting Building Parts

The results show that the parameters used for the selected building parts depend on the use of concrete materials in non-structural works during the architecture phase and the same concrete materials' strength not compromising the part's function or description, with these considerations also applying to building elements. For example, non-reinforced

concrete was applied in internal parts, such as the secrete, lintels, and lift wall, and in external parts, such as the carp stone, flowerbeds, soft landscape, walk area, pavement, wall boundary panel, entrance steps, and printed concrete. Figure 4 shows the final selection of non-structural building parts. On the other hand, reinforced concrete was applied in the production of beams, columns, suspended slabs, and staircases in all of the building references without changing the original and modified buildings.

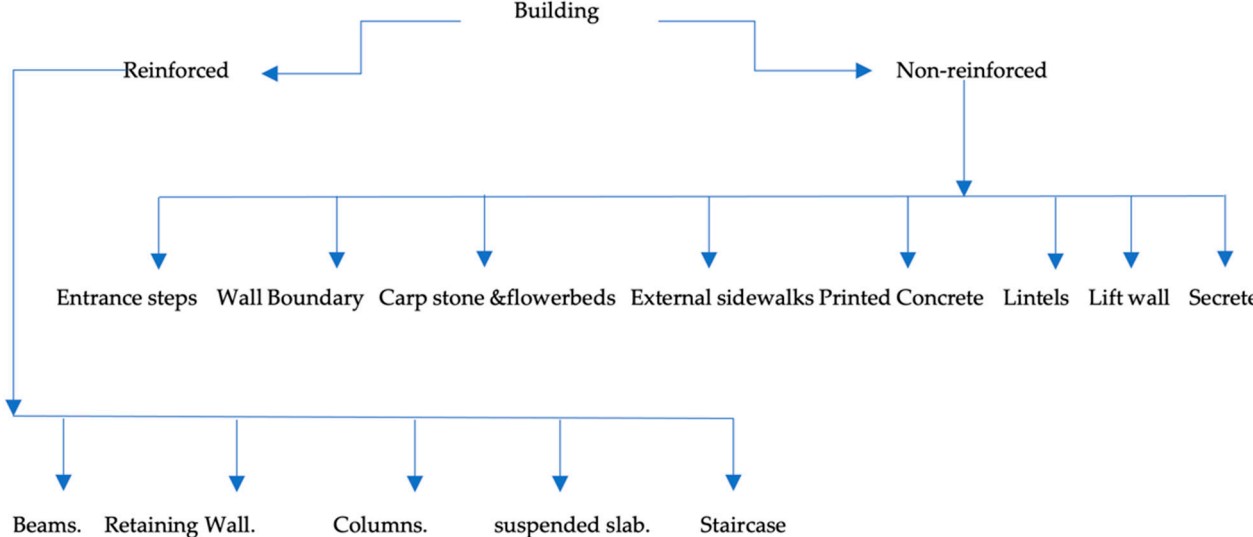

**Figure 4.** The selection of the structural and non-structural building parts.

### 3.3. Classifying the Concrete Materials

In this research, concrete materials are classified into three categories depending on their functionality and durability as follows:

The first type is C1, namely reinforced concrete using OPC, which is considered one of the most important and common construction materials worldwide and is usually used for general uses in Egypt. For example, according to the current residential building project, it is used for panels, beams, foundations, suspended slabs, and staircases.

The second type is C2, which is a non-reinforced concrete with OPC used for different, fundamental construction works, such as entrance steps, wall boundary panels, carp stones, flowerbeds, soft landscapes, walkways, printed concrete, lintels, lift walls, and secretes.

The third type is C3, which is a non-reinforced RAC made from recycled aggregate obtained from C&D waste. However, whereas the selection of C1 and C2 concrete material is based on the data collected from official building documents, C3 selection depends on sustainability features and the maintenance of durability, strength, and workability in the substitution concrete under the Egyptian building codes and Egyptian concrete codes.

Furthermore, different mix concrete proportions were used, such as 1:4:8 (M10) and 1:2:4 (M15) for non-structural selected building parts and 1:1:2 (M25) for structural parts such as beams, columns, and foundations, using the absolute volume method, which considers the volume of the concrete mix to be equal to each component's absolute volume, namely cement, sand, RCA, and water. The volume of a rectangular cross-sectional building part can be calculated as length x width x height. We calculated the CC quantity based on the bill of quantity (BOQ) and then estimated the GC quantity for each reference using Microsoft Excel to obtain hypothetical options with the same design features, with equivalent functionality and durability levels for non-structural works in the selected parts. In addition, we calculated the cement, natural aggregate, and sand consumption for each reference, respectively.

Concrete Materials' Quantities for Specific Building Parts

Table 2 shows the C1 and C2 quantities for original A and modified A in the selected building parts. This represents the first attempt to reduce concrete consumption by means of design modification before integrating GC, and we noticed a reduction in CC quantities by 8.14% from original A to modified A. Furthermore, the relationship between the selected building parts and (C1 and C2) quantities was identified. The results show an increase in the concrete quantity in modified A compared with original A, due to an increase in the internal floor area after the design modification, as shown in Figure 5. For more details regarding the selected building parts, concrete, and cement quantities, see Table A1 in the Appendix A.

**Table 2.** Concrete materials C1 and C2's quantities in different selected parts in the original and modified building designs.

| Concrete Specification | Building Part | Concrete with Ordinary Portland Cement | |
|---|---|---|---|
| | | Original Building A C1 and C2, m³ | Modified Building A C1 and C2, m³ |
| Non- structural concrete | Internal floor (secrete) | 47.77 | 52.89 |
| | Wall boundary panel | 26.46 | 26.46 |
| | External sidewalk, carp stone, printed concrete | 38.85 | 38.85 |
| | Internal walls (lintels and lift walls) | 37.49 | 36.79 |
| | Entrance steps | 3.60 | 3.60 |
| Structural concrete | | 347.34 | 302.09 |
| | Total concrete | 501.51 | 460.68 |
| | Percentage | 100% | 91.9% |

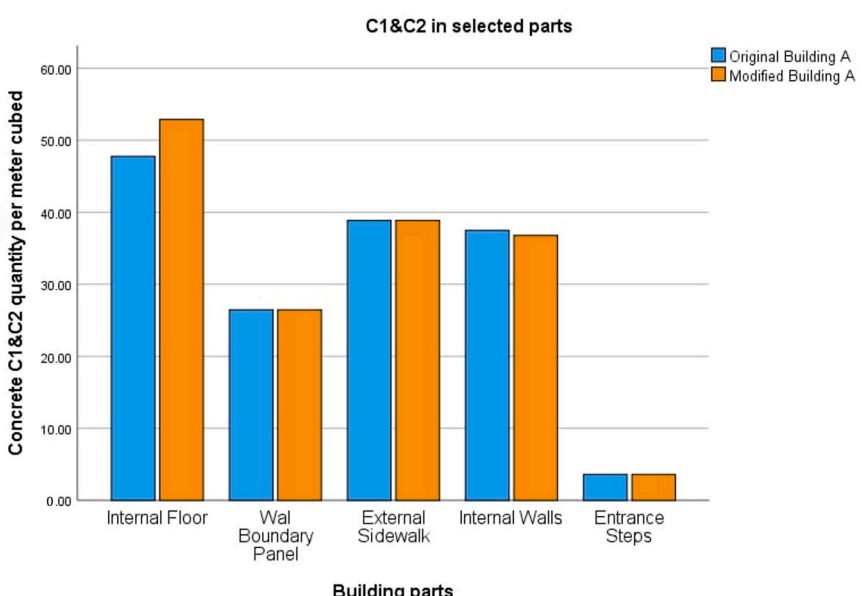

**Figure 5.** The relationship between selected building parts and C1 and C2 concrete materials in original A and modified A.

### 3.4. Applying Concrete Materials in the Selected Building Parts

As seen in Table 2, a reduction in the quantity of concrete materials from original A, (C1 and C2), at 347.34 and 154.18 m³, respectively, to modified B (C1 and C3), at 302.09

and 158.95 m$^3$, respectively, could be observed. On the other hand, the cement quantity in original A (C1 and C2) was 189,053 and 47,949 kg, respectively, while in Modified B (C1 and C3) it was 164,424 and 26,560 kg, respectively, achieving the greatest reduction in concrete and cement consumption. Identifying the Cement, Natural Aggregate, and Sand for Each Reference. Table 3 illustrates the results of calculating the consumption of materials when using concrete materials C1, C2, and C3 in the studied buildings (original A, original B, modified A, and modified B), respectively. The results show that modified B, which contains the GC material C3, is considered the most optimal version, with its cement consumption being 190,984 kg, representing 80.6% of the total, and there being no significant differences between original B, the cement consumption of which is 214,872 kg, and modified building A, for which this value is 213,749 kg. Original A is the biggest consumer of cement, with a total of 237,002 kg. On the other hand, the quantities of sand, natural coarse aggregate, and RCA show a significant reduction.

**Table 3.** The quantities of cement, sand and aggregate in C1, C2, C3 concrete types in original A, original B, modified A, and modified B, respectively.

| | Original Building A and B and Modified Building A and B | | | | | | | | Notes |
|---|---|---|---|---|---|---|---|---|---|
| | **Original A** | | **Original B** | | **Modified A** | | **Modified B** | | |
| Total Concrete, m$^3$ | 501.51 | | 501.51 | | 460.68 | | 460.68 | | |
| | C1 | C2 | C1 | C3 | C1 | C2 | C1 | C3 | |
| Concrete, m$^3$ | 347.34 | 154.18 | 347.34 | 154.18 | 302.09 | 158.59 | 302.09 | 158.59 | |
| | Cement, Sand, Natural Aggregate, and RCA | | | | | | | | |
| Code Specification | CI | C2 | CI | C3 | CI | C2 | CI | C3 | |
| | 1:1:2 | 1:2:4 | 1:1:2 | 1:4:8 | 1:1:2 | 1:2:4 | 1:1:2 | 1:4:8 | |
| Cement, kg | 189,053 | 47,949 | 189,053 | 25,818 | 164,424 | 49,325 | 164,424 | 26,560 | Cement from OPC |
| | 237,002 | | 214,872 | | 213,749 | | 190,984 | | |
| Cement, % | 100% | | 90.1% | | 90.2% | | 80.6% | | |
| Sand, kg | 410,246 | 208,096 | 410,246 | 112,052 | 356,801 | 214,069 | 356,801 | 115,268 | |
| | 618,342 | | 522,298 | | 570,870 | | 472,069 | | |
| Sand, % | 100% | | 84.5% | | 92.3% | | 76.3% | | |
| NA kg | 714,621 | 362,490 | 714,621 | – | 621,524 | 372,895 | 621,524 | — | |
| | 1,077,119 | | 714,621 | | 994,419 | | 621,524 | | |
| | 100% | | 66.3% | | 92.3% | | 57.7% | | |
| RAC, kg | — | — | — | 195,187 | — | — | — | 200,790 | |
| | — | | 195,187 | | — | | 200,790 | | |
| NA + RAC, kg | 1,077,119 | | 909,809 | | 994,419 | | 822,314 | | |
| Aggregate, % | 100% | | 84.5% | | 92.3% | | 76.3% | | |

### 3.5. The Selection of the Most Optimum Reference

Modified B was considered a significant building reference in the context of reducing materials and was selected as the optimal reference for cement, sand, NA, and RCA, as shown in Table 4.

**Table 4.** The quantities of C1 and C3 concrete in modified building B.

| C1 | Concrete Component C1, Modified Building B | | |
| --- | --- | --- | --- |
| | Cement | Sand | Aggregate |
| Mix/Ratio | 1 | 1 | 2/NCA |
| Qt/kg | 164,424 | 356,801 | 621,524 |
| C3 | Concrete Component C3, Modified Building B | | |
| Mix/Ratio | 1 | 4 | 8/RCA |
| kg | 26,560 | 115,268 | 200,790 |

The process of reducing cement materials is represented in Table 5 in three steps: first, the reduction achieved after original A was modified to original B was 9.3%; second, the reduction due to applying GC and via modified A and modified B was 10.7%; and third, the final reduction score between modified B and original A was 19.4%, a positive indicator of achieving the research goal.

**Table 5.** The reduction in concrete and cement quantities through the three steps of the modified process.

| | First | | | | |
| --- | --- | --- | --- | --- | --- |
| Description | Original Building | | | | Notes |
| | Original A | | Original B | | |
| Concrete type | C1 | C2 | C1 | C3 | |
| Concrete type/m$^3$ | 347.34 | 154.16 | 347.342 | 154.16 | |
| Total concrete, m$^3$ | 501.507 | | 501.507 | | Total saving of cement (OPC) in the project. |
| Cement, kg | 189,053 | 47,948 | 189,053 | 25,818 | |
| Total cement, kg | 237,002 | | 214,872 | | |
| Cements, % | 100% | | 90.7% | | 9.3% |
| | Second | | | | |
| Description | Modified Building | | | | Notes |
| | Modified A | | Modified B | | |
| Concrete type | C1 | C2 | C1 | C3 | |
| Concrete type, m$^3$ | 302.09 | 158.59 | 302.09 | 158.59 | |
| Total concrete, m$^3$ | 460.68 | | 460.68 | | Total saving of Cement (OPC) in the project. |
| Cement, kg | 164,424 | 49,325 | 164,424 | 26,559 | |
| Total cement, kg | 213,749 | | 190,984 | | |
| Cements, % | 100% | | 89.4% | | 10.6% |
| | Third | | | | |
| Description | Original Building A and Modified Building B | | | | Notes |
| | Original A | | Modified B | | |
| Concrete type | C1 | C2 | C1 | C3 | |
| Concrete type, m$^3$ | 347.34 | 154.18 | 302.09 | 158.59 | |
| | 501.51 | | 460.68 | | Total saving of cement (OPC) in the project. |
| Total concrete, m$^3$ | | | | | |
| Cement, kg | 189,053 | 47,948 | 164,424 | 26,559 | |
| Total cement, kg | 237,002 | | 190,984 | | |
| Cement% | 100% | | 80.6% | | 19.4% |

### 3.6. Comparison Analysis of CC and GC

A comparative analysis of the quantities of CC and GC materials in all references and their components is shown in Figure 6, presenting the relationship between the CC quantity and its components before adding GC.

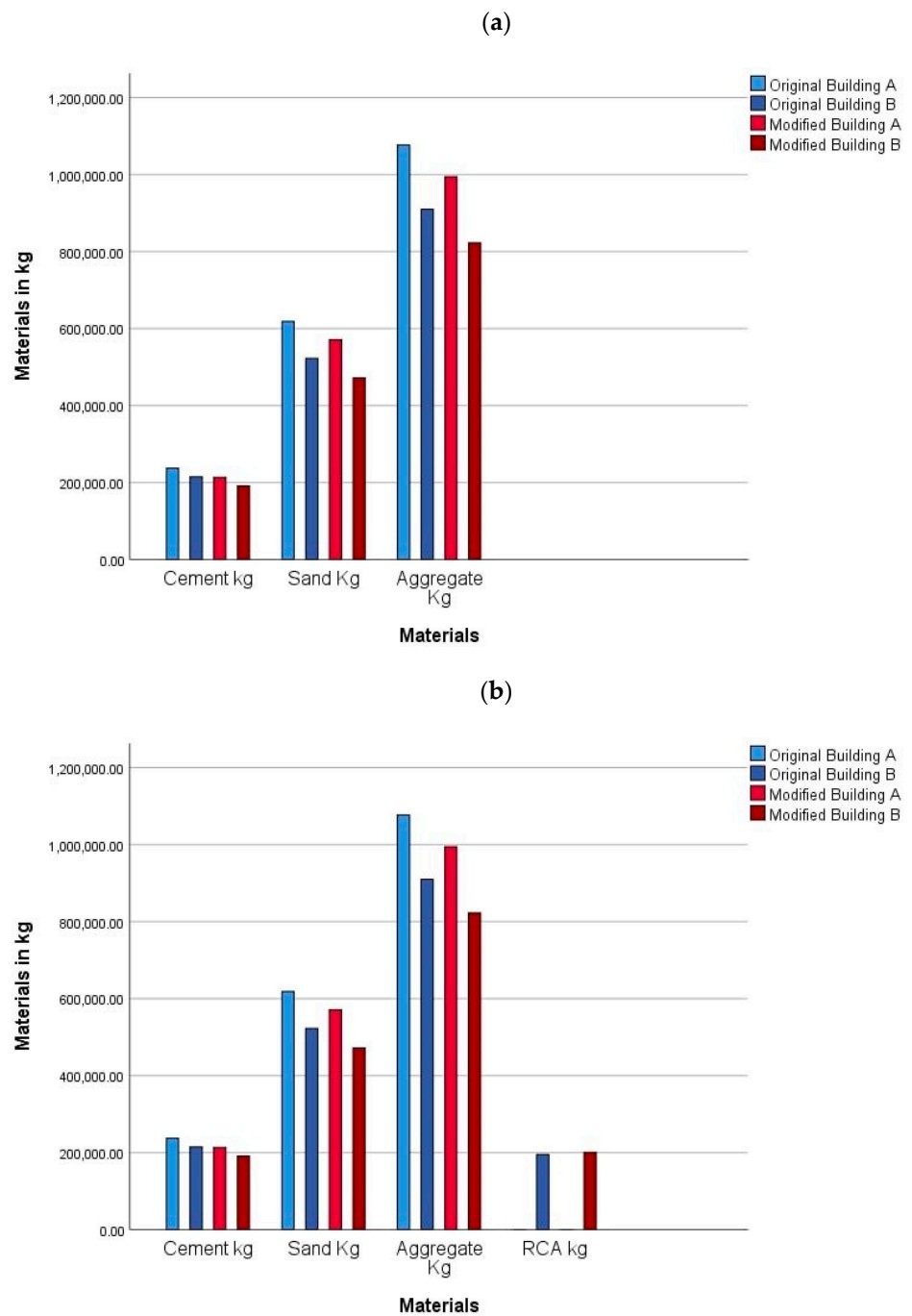

**Figure 6.** (**a**) The relationship between the quantities of CC components in all references; (**b**) the relationship between the CC components and RCA quantity in all references.

Figure 7 presents the relationship between the consumption of different materials after using GC with RCA, and shows that modified B provides the optimal reduction in cement, sand, and NA consumption.

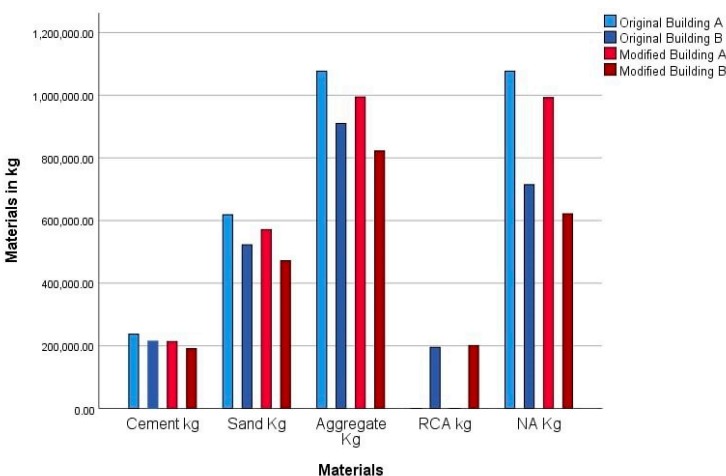

**Figure 7.** The relationship between the quantity of CC components, RCA, and NA in all references.

*3.7. Cement Production and $CO_2$ Emissions*

In this research, $CO_2$ emissions were estimated from cement production, assuming that cement is the OPC type, and the cement quantity per ton was calculated by using the CDIAC's method [43,44]. Thus, emissions estimates from cement production were as follows: Original A cement consumption = 236.9 tons, which produced 118.45 $tCO_2$.

Original B cement consumption = 214.8 tons, which produced 107.4 $tCO_2$.

Modified A cement consumption = 213.7 tons, which produced 106.85 $tCO_2$.

Modified B cement consumption = 191.0 tons, which produced 95.5 $tCO_2$.

Therefore, the reduction in $CO_2$ emissions from original A to modified B was 22.95 t, which represents a 19.4% reduction. However, the amount of cement consumption/ton and reduction percentage is shown in Table 6, which summarizes the relationship between the original and modified building references in terms of structural and non-structural concrete. The research was concerned with investigating the cement consumption in the specific selected building parts, and the results show that modified B contributed to a reduction in cement consumption of 19.5% from original building A. In addition, cement consumption was reduced by 44% in non-structural concrete.

**Table 6.** The cement consumption per ton and cement reduction % for each original and modified reference, respectively.

| | Original Building A and B and Modified Building A and B | | | | | | |
|---|---|---|---|---|---|---|---|
| | **Original A** | | **Original B** | | **Modified A** | | **Modified B** |
| Concrete, m$^3$ | 501.51 | | 501.501 | | 460.68 | | 460.68 |
| Mix type | CI | C2 | CI | C3 | CI | C2 | CI | C3 |
| Cement, ton | 189.0 | 47.9 | 189.0 | 25.9 | 164.4 | 49.3 | 164.4 | 26.6 |
| Cement, ton | 236.9 | | 214.8 | | 213.7 | | 191.0 |
| Cement C1, C2 and C3 % | 100% | | 90.07% | | 90.2% | | 80.6% |
| Cement C1, C2 and C3 Reduction % | 100% | | −9.3% | | −9.8% | | −19.5% |
| Cement C2 and C3, ton | 47.9 | | 25.9 | | 49.3 | | 26.6 |
| Cement C2 and C3 % | 100% | | 53.8% | | 102% | | 55.5% |
| Cement C2 and C3 reduction % | 100% | | −46.2 | | +2.9% | | −44.5% |

Clarifying the relationship between cement consumption and reduction illustrated in Figure 8, we can observe a clear relationship between the building references and the final cement consumption. Therefore, modified building B, considered the best reference for reducing the total cement consumption, reduces $CO_2$ emissions by 44.5% compared with original A. However, the investigation of the current research was concerned with

reducing the environmental impact of non-structural concrete works in selected building parts by reducing the cement's content of natural aggregates. On the other hand, using an integration method by applying modified, selected RCA produced locally from C&D waste as well as selected building parts, which, in most cases, have been neglected in scholarly investigations, can play a vital role in protecting the environment and enhancing the construction sector at the local and international level.

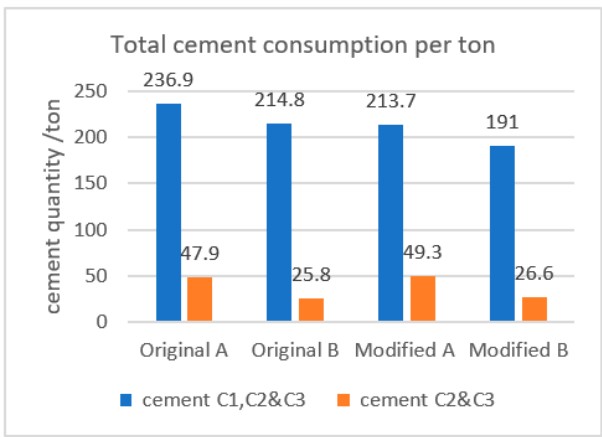 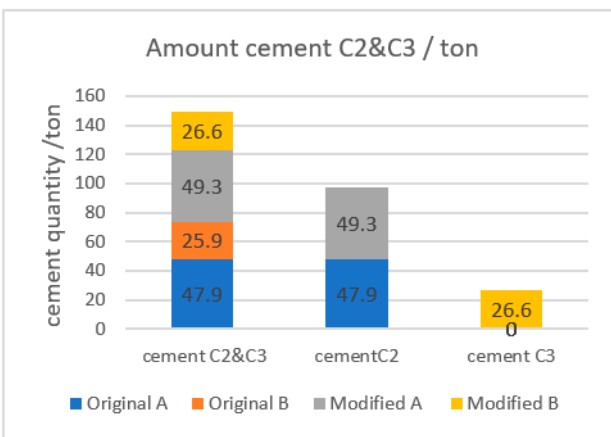

**Figure 8.** The relationship between cement consumption and reduction in the different building references.

In this regard, the production of OPC is required for the four references, original A, original B, modified A, and modified, B, respectively. The graph classifies cement into two groups: first, the amount of cement in all concrete types, C1, C2, and C3, considering structural and non-structural works for each building. Second, the amount of cement in C2 and C3, consequently considering non-structural works for each building reference. However, modified building A was the reference with the highest cement consumption, which indicates that the strategy used for modification should be applied in the same order and context to obtain the most optimal result.

### 3.8. Sustainable Building Model

The 3D model, the modified B model, was developed to consider a clear image in the conceptual design stage framework. Figure 9 shows the concept design framework and its contribution to material consumption and environmental protection. We started by redesigning and modifying the original design to develop a conceptual reference able to substitute CC with GC in specific selected building parts. The reused RCA came from C&D materials to reduce the consumption of OPC and NA and to reduce $CO_2$ emissions. Therefore, the final optimal model is presented as a guideline for stakeholders, aiding in the decision-making of the authorities governing the residential building sector in Egypt. Furthermore, the role of the design phase in the residential building sector is presented, reflecting its positive impact on applying sustainable construction on a wider scale all over the country.

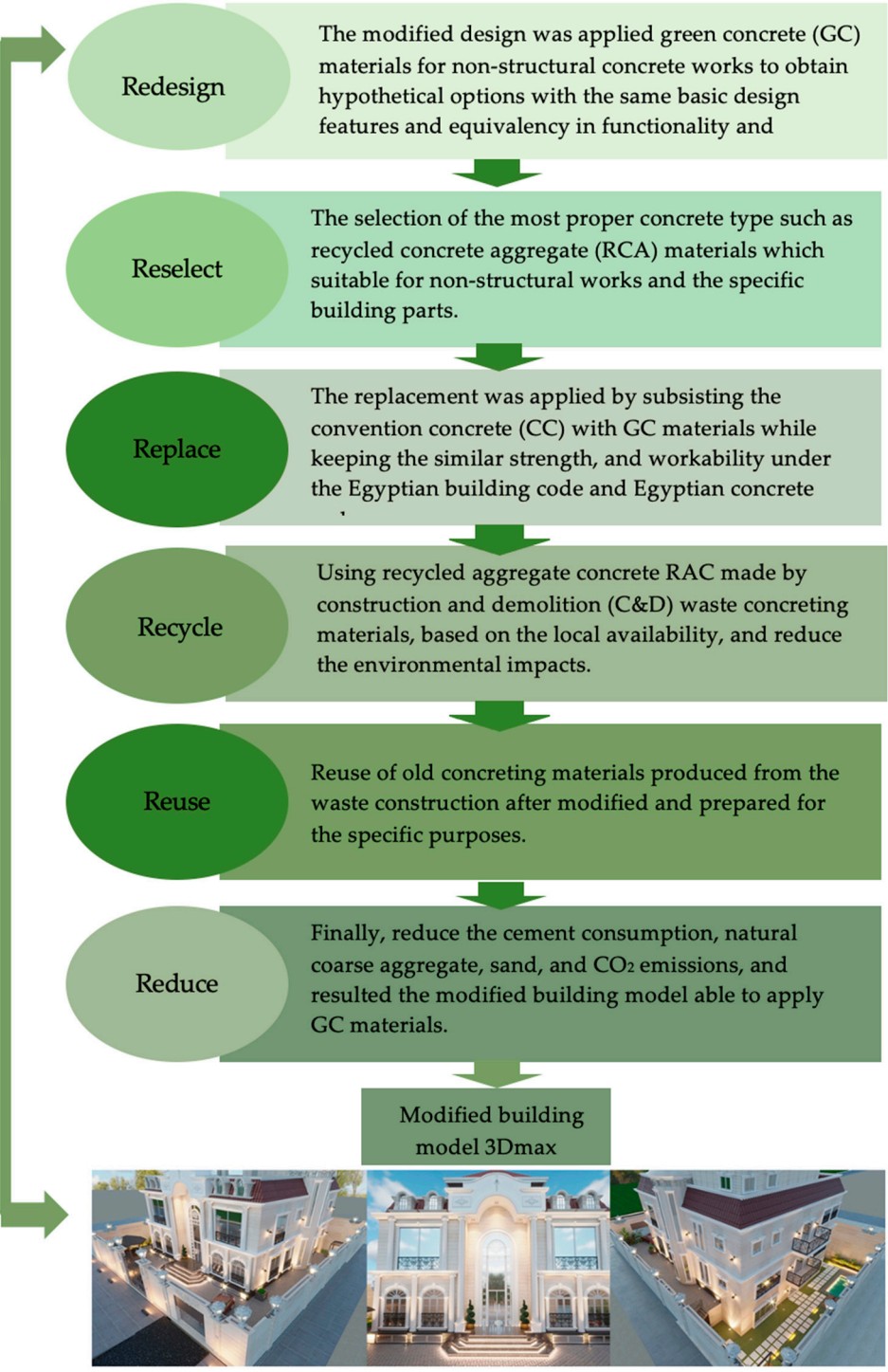

**Figure 9.** The conceptual design framework of applying GC materials to the modified B 3D model.

## 4. Discussion

The current study investigated and analyzed the effects of substituting CC materials with GC using original and modified designs with different concrete types, C1, C2, and C3, and identified the resulting reductions in cement, sand, the consumption of natural coarse aggregates, and $CO_2$ emissions. Thereby, DBS was used as a sustainable integrated method, which included redesigning, re-selecting, replacement, recycling, reusing, and reducing strategies and defined effective methods for achieving sustainability in the construction materials sector.

The redesign stage therefore applied modifications to the original building design to make it more suitable for the replacement of CC materials with GC. This stage confirmed a reduction in OPC from original A and original B of 9.3%. Accordingly, the reduction reached 19.4% after the reselection of non-structural building parts and applying GC in modified B. As a result, the study proved that the modification of the building design and the suitable selection of building parts added extra value and played a vital role in reducing the consumption of cement materials in the stages of design and construction [35,47]. Furthermore, the criteria for substituting GC materials in the selected building parts were achieved, thereby maintaining a functionality and durability equivalent to CC materials without compromising the material's descriptions and applications based on Egyptian building code considerations. Therefore, the replacement of C2 with C3, which contain OPC and RAC, respectively, in non-structural building parts played a vital role and improved the strength and durability of recycled concrete materials, and this is reflected in previous research efforts in the same context [2,5,21,42]. On the other hand, the recycling stage depended on the use of C&D waste, and RAC is considered as a suitable concrete material made with RCA due to its durability, availability, and strength. GC materials enhance durability and can reduce environmental impacts [48]. Adding more GC reduced the construction waste stream at the local level of the Egyptian building sector, and accordingly reduced any negative environmental impacts [33].

Furthermore, the reuse of RAC in original B and modified B as a partial replacement, with consideration given to the design mix proportion of the concrete materials, achieved the greatest reduction in total concrete consumption compared with original A, and this is consequently reflected in the total quantity of cement, sand, and NCA, and confirmed by [21,26,30,31]. In this study, the application of GC in internal and external non-structural works such as secretes, printed concrete, carp stones, pavements, and sidewalks was addressed and identified, which proved its ability to enhance concrete properties and strength [2,10,21,25,31,37,49].

Therefore, the consumption of raw construction materials can be reduced by using sustainable concrete made from recycled materials. In the same context, the research showed a reduction from 618.3 to 472.1 tons in the consumption of sand between the original A and modified B references. Furthermore, due to the equivalent mix proportion ratio, a reduction in the consumption of natural coarse aggregate from 1,077,119 to 621,524 kg, by approximately 23.7%, was observed for both materials. Another reason for using GC can be reduced $CO_2$ emissions; therefore, the replacement of CC with GC addresses environmental concerns related to the high-level consumption of fossil fuels and the depletion of raw materials [48].

## 5. Conclusions

Sustainable construction materials reduce negative environmental impacts, especially $CO_2$ emissions and the consumption of natural resource. In the last few decades, the fast rate of population growth in Egypt has led to an increase in the number of residential buildings all over the country, which is associated with a massive increase in construction material consumption, especially concrete materials.

The OPC is considered the most common construction material locally and globally and is considered one of the largest producers of $CO_2$ during its production process. Therefore, there is a critical necessity for demonstrating alternative innovative green solutions, such as GC materials, to meet the needs of the construction industry at present without compromising natural resources for future construction projects. The use of GC materials was addressed based on the applied DBS strategy in a residential building case study. It began with redesign, reselection, replacement, recycling, and reuse strategies, and resulted with a significant reduction in environmental burdens. In addition, the research presented sustainable integrated methods, which are considered a novel and effective strategy for reducing cement and natural aggregate consumption, leading to a reduction in environmental impacts and the achievement of sustainability in the construction materials sector.

The original design was modified, and we presented a sustainable model that can replace CC with GC in non-structural works in specific building parts. Using GC materials based on actual measurements and accurate data calculations presented strong evidence for the implementation of GC materials in the residential building sector and can be successfully applied to improve the sustainable features of buildings. Furthermore, we analyzed and compared the substitution of CC materials with GC, using original and modified design scenarios with different concrete types, which enhanced the practical use of GC in order to help ensure a long-term environmental solution for current and future construction projects. This study used GC materials in an actual family residential building in New Borg El-Arab, Egypt. The building is considered a family contemporary residential villa and is one of the most common building types in new Egyptian cities all over the country. The sustainable building design model was developed and applied to achieve a broad impact for applying sustainable construction materials such as RAC, instead of OPC, in non-structural works. However, it can be used in many countries worldwide under the same building codes and concrete material regulations.

On the one hand, using RCA made from C&D waste played an essential role in reducing $CO_2$ emissions and eliminating the depletion of natural resources; on the other hand, it promoted a positive effect by limiting the waste construction stream, reducing air pollution, and minimizing the total construction waste amount sent to landfill. Therefore, the criteria for the selection of RAC materials were based on the availability of materials and equivalent durability, strength, and functionality parameters. Consequently, the study results significantly reduced cement, sand, and NCA consumption after substituting CC materials with GC in the selected building parts. For example, modified B reduced the cement consumption by 19.5%, compared with original building A, and we observed a 44% reduction for non-structural concrete in the early design stage.

The study proved that using GC instead of CC materials can reduce cement and natural aggregate consumption and $CO_2$ emissions, adding more environmental benefits from recycled materials made from C&D waste. The replacement of CC should be considered during the early architectural design stages, with cooperation between all of the construction parties, and the regulations for using GC materials in the residential sector at a wider scale need to be improved in order to further reduce negative environmental impacts. This study presents a helpful guideline for designers, contractors, stakeholders, authorities, decision-makers, and those responsible for the construction and concrete material industries. Finally, based on the results of this study, it is expected that GC materials would yield sustainable construction activities in the residential building sector worldwide.

**Author Contributions:** Conceptualization, H.M., G.K. and G.S.; methodology, H.M. and G.S.; software, H.M. and G.S.; validation, H.M., G.S. and G.K.; formal analysis, H.M. and G.S.; investigation H.M., G.S. and G.K.; resources, H.M., G.S. and G.K.; data curation, H.M. and G.S.; writing—original draft preparation, H.M.; writing—review and editing, H.M., G.S. and G.K.; supervision, G.S. and G.K. All authors have read and agreed to the published version of the manuscript.

**Funding:** This research received no external funding.

**Institutional Review Board Statement:** Not applicable.

**Informed Consent Statement:** Not applicable.

**Data Availability Statement:** Data produced and processed in this study are included in the published article. In addition, datasets can be acquired from the corresponding author for appropriate purposes.

**Acknowledgments:** We appreciate the support given by Project No. TKP2021-NKTA-32, which has been implemented with the support provided by the National Research, Development, and Innovation Fund of Hungary, financed under the TKP2021-NKTA funding scheme.

**Conflicts of Interest:** The authors declare no conflict of interest.

## Abbreviations

| | |
|---|---|
| OPC | Ordinary Portland Cement |
| GC | Green Concrete |
| CC | Conventional Concrete |
| RAC | Recycled Aggregate Concrete |
| RCA | Recycled Concrete Aggregates |
| DBS | Design-Based Solution |
| NCA | Natural Coarse Aggregates |
| BOQ | Bill Of Quantity |
| C&D | Construction and Demolition |

## Appendix A

**Table A1.** The quantities of non-reinforced concrete and cement in original B with C2 and modified B with C3 in the selected building parts.

| | Non-Structural Concrete in Original Building B (C2) and Modified Building B (C3) in the Selected Building Parts | | | | | | | | | | | | |
|---|---|---|---|---|---|---|---|---|---|---|---|---|---|
| | Internal Floor | | | | Wall Boundary | External Floor | | | | Internal Walls | | | Entrance |
| | Basement | Ground Floor | First Floor | Penthouse | Garden | Garden Printed Concrete | Pavement Carp Stone | Sidewalks | Basement | Ground Floor | First Floor | Penthouse | Ground Floor |
| OB | 12.1 | 16.9 | 15.2 | 3.5 | 26.5 | 31.1 | 6.7 | 0.96 | 9.5 | 10.4 | 11.1 | 6.4 | 3.6 |
| Total, m³ | 47.8 | | | | 26.5 | 38.9 | | | 37.5 | | | | 3.6 |
| MB | 14.3 | 18.2 | 16.5 | 3.9 | 26.5 | 31.1 | 6.8 | 0.96 | 9.3 | 10.3 | 10.9 | 6.2 | 3.6 |
| Total, m³ | 52.89 | | | | 26.46 | 38.85 | | | 36.79 | | | | 3.6 |
| | | | | | | | | | Total concrete/m³ | | | Total cement/kg | |
| Total concrete, Original Building A | | | | | | | | | 154.17 | | | — | |
| Total cement, C2 | | | | | | | | | — | | | 47,949 | |
| Total concrete, Modified Building A | | | | | | | | | 158.60 | | | — | |
| Total cement, C2 | | | | | | | | | — | | | 49,325 | |
| Total concrete, Original Building B | | | | | | | | | 154.16 | | | — | |
| Total cement, C3 | | | | | | | | | — | | | 25,818 kg | |
| Total concrete, Modified Building B | | | | | | | | | 158.6 | | | — | |
| Total cement, C3 | | | | | | | | | — | | | 26,559 kg | |

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
