# Peer review of "Effects of Using Green Concrete Materials on the CO2 Emissions of the Residential Building Sector in Egypt"

_sustainability, doi:10.3390/su14063592_

Round 1

Reviewer 1 Report

The goal of this study is to see how replacing green concrete for conventional concrete in non-structural concrete works of a residential structure in New Borg El-Arab, Egypt, affects the results. This study is new and interesting, but the following comments should be carefully addressed by authors:

  • The paper should be extensively improved in terms of English particularly fluency.
  • In abstract section, Natural Coarse Aggregate should be replaced by natural coarse aggregate. Please check all capitals over the manuscript.
  • All figures should be improved. The captions and numbers used in figures are so small and unreadable.
  • The novelty of this study should be extensively highlighted.
  • The reason for selected cases study should be highlighted. What was the main reason for selecting the mentioned cases? Are they comparable?
  • Cost analysis would be useful and interesting
  • Vertical axis of Figure 8 should be labeled.
  • Flowchart presented in Figure 9 is not understandable. Please replace it with more understandable flowchart
  • The conclusion section has been written very weak. So, this section should be extensively re-written.
  • Avoid from presenting Arabic language in the paper. For example, reference [31]

Considering the presented comments above, the paper should be extensively improved and review for another revision.

Author Response

Dear Reviewer,

Thank you so much for the valuable comments. Please find the following modifications:

Comments:

1.      

The paper should be extensively improved in terms of English particularly fluency.

 We sent the manuscript to MDPI author's services experts for reviewing  the English and improving the fluency by native speakers’ experts.

2.      

In the abstract section, Natural Coarse Aggregate should be replaced by natural coarse aggregate. Please check all capitals over the manuscript.

The capital letters checked and corrected entire all the document

3.      

All figures should be improved. The captions and numbers used in figures are so small and unreadable.

 All the figures in the manuscript were reviewed and improved in terms of size and readability.

4.      

The novelty of this study should be extensively highlighted.

 We agree that the research novelty should be addressed in more detail. So, it highlighted through the objective section, that several previous studies discussed the properties, strength, durability of GC materials. However, there is a limitation in the studies concerned with applying the green concrete in the specific building parts in non-structural works for the residential building sector.

Therefore, the research considers the first attempt to apply green concrete and presents the first sustainable building model using GC materials in the residential buildings sector in Egypt.

5.      

The reason for selected cases study should be highlighted. What was the main reason for selecting the mentioned cases? Are they comparable?

The reasons for the selection of the case study and added that:

· The case study considers a common residential building type, especially in the new Egyptian cities. The number of will expected to increase in the future as part of the government development strategy; to face the population growth crisis, we indicate that the research will achieve significant results and it will be helpful in the building and environmental authorities.

· Applying the GC materials in such building types in Egypt can be used in any country. It will encourage the construction companies, architects, engineers, and the concrete industry to apply sustainable construction materials in future projects.

· Furthermore, the building was chosen depending on specific details and data; it is difficult to access such detailed data, but we have owner permission and approval from all the responsible parties, and the ability to have precise data present valuable results.

· Moreover, the permission for in situ measurements, visiting the site, and discussion with construction companies' expertise increases the chance for using green concrete on a wide scale and leads to establishing new building rules and regulations in the future.

The main reason for selecting the research case study:

· Increase the use of green building materials and reduce the negative environmental impacts for conventional concrete materials. by applying them to the family residential building sector, aiming to close the gap between the academic and building industry field and focus on the importance of sustainable construction materials.

Are they comparable?

· Yes, they have a strong chance to apply different comparative aspects with other building types locally and globally under the same building and concrete codes.

6.      

Cost analysis would be useful and interesting

 We agree that cost analysis has an essential role. It will be a valuable and interesting investigation; based on that, we prepared to identify it in detail in a separate study because it still needs more collecting data about the concrete market price. So, we do not want to deal with this topic in this article because it is not under the current article scope.

7.      

Vertical axis of Figure 8 should be labeled.

The vertical axis applied

8.      

Flowchart presented in Figure 9 is not understandable. Please replace it with more understandable flowchart

 The flowchart design changed, and explain each step in detail to be more understandable for the reader.

9.      

The conclusion section has been written very weak. So, this section should be extensively re-written.

 The conclusion chapter changed and improved.

It applied more details to summarize all the study chapters and sections to increase its effectiveness and impact.

10.   

Avoid from presenting Arabic language in the paper. For example, reference [31]

 Corrected and apply by English.

Kind regards.

Authors

Reviewer 2 Report

This paper presents a summary of the effect of substituting the conventional concrete (CC) material with green concrete (GC) of non-structural concrete works of a residential building. The topic addressed is interesting and deserves a constructive discussion for reducing environmental impact and achieving sustainability in building sector. However, I think that there are a few improvements that should be made before publication.

1.p.3 Line124,p.4, Line176, p.5 Line239

Please change the section number correctly.

2. p.11 Table2

“Modified Building A” in Table 2 replace by “Modified Building B”.

The text describing the table does not match the contents of the table. Please confirm.

3. p.11 Section 3.4

Please provide the relevant table number in the text.

4. p.13 Line 551-553

Please add a description so that it is clear that the text is related to Figure 7.

5. p.16 Line661, p.17 Line707

Please change the chapter number correctly.

Author Response

Review 2

Dear Reviewer,

Thank you so much for the valuable comments. Please find the following modifications:

Comments:

1.

Regarding the 1.p.3 Line124,p.4, Line176, p.5 Line239, please change the section number correctly.

The section number checked and change

2.

p.11 Table2 “Modified Building A” in Table 2 replace by “Modified Building , The text describing the table does not match the contents of the table. Please confirm

 we agree that it’s essential to match the text with the entire table, so the text is revised and improved to be more matching, and the correction applied to the text by change (modified B )by (modified A).

3.

p.11 Section 3.4 Please provide the relevant table number in the text.

The table2 number is provided in section 3.4. and checked all the tables number in the entire document.

4.

p.13 Line 551-553 Please add a description so that it is clear that the text is related to Figure 7.

Figure 7 described in the text as a following “As well as figure 7, presents the relationship between all the materials consumption after using GC with RCA, and showed that the Modified B has the most optimum reduction in cement, sand, and NA.” and moved before the figure directly to improve the readability.

5.

p.16 Line661, p.17 Line707 Please change the chapter number correctly.

The chapter number was revised and checked correctly

 Kind regards,

 Authors

Round 2

Reviewer 1 Report

The reviewer would like to appreciate authors to address all comments carefully. The paper is recommended for publication in the current format